# Combining Eco-Design and LCA as Decision-Making Process to Prevent Plastics in Packaging Application

**Eleonora Foschi** [1,*], **Sara Zanni** [1] **and Alessandra Bonoli** [2]

1   Department of Management, University of Bologna, 40126 Bologna, Italy; sara.zanni7@unibo.it
2   Department of Civil, Chemical, Environmental and Materials Engineering, University of Bologna, 40131 Bologna, Italy; alessandra.bonoli@unibo.it
*   Correspondence: eleonora.foschi3@unibo.it

**Abstract:** The diffusion of the culture of sustainability and circular economy increasingly pushes companies to adopt green strategies and integrate circular business models in the corporate agenda. It assumes higher relevance in the packaging industry because of the growing plastics demand, the increasing awareness of consumers on single-use-products, the low recyclability performance and last but not least, the challenge of urban littering and microplastics dispersion in marine ecosystem. This paper presents the case of a small-medium enterprise that implemented a decision-making process to rethink the design of frozen food packaging in accordance with systemic and life cycle thinking. Eco-design and Life Cycle Assessment (LCA) have been simultaneously used to test and validate the redesign process, thus fostering the substitution of the plastic "open and close" cap with a closing method entirely made of cardboard. Results shows how using an integrated decision-making system at the design stage have allowed to get up many benefits at multiple levels, including sustainable and safe supply chain, efficient logistic operations, better recyclability, and lower energy consumption. Moreover, even if it cannot be assessed by the existing tools, the solution provides a strong contribution to the reduction in the consumption of plastics and the prevention of marine pollution.

**Keywords:** eco-design; plastic prevention; packaging; sustainability; LCA

## 1. Introduction

Among all the materials, plastics have shown the fastest growing demand in the last decades, moving from 1.7 million metric tons (Mt) in 1950 to 359 Mt in 2018 [1–5]. Its negligent use, especially in the packaging sector, has caused multiple externalities over the years at such a point to become one of the most urgent environmental issues to deal with nowadays. Despite significant worldwide advances in management, treatment and recycling of plastics in the last three decades, the largest amount of post-consumer waste properly collected is incinerated and/or landfilled [5,6]. In some countries, a residual part is openly burned, emitting carbon monoxide (CO) and carbon dioxide ($CO_2$) [7]. Illegal export and uncontrolled disposal are also frequent because of the underperforming recycling infrastructure [8,9]. Finally, an unknown portion of plastics is littered, thus contributing to massively worsen the status quo of the marine ecosystem [10,11]. The literature on marine plastic pollution (MPP) leaves no doubt that plastics make-up most of the marine litter worldwide. As reported by Derraik and Thevenon et al., 60–80% of global marine debris is made of plastic materials [12,13]. Even if standardized methods to measure and model the presence of plastics and microplastics in seas and beaches doesn't exist yet, the key publications on the topic estimates that about 10% of the total plastics produced every year is released in the oceans, accounting for 10 Mt [14–16]. Owing to the low degradability, conventional plastics remain in aquatic habits for decades [17]. Recent sampling

activities reveals that many commercial packets and wrappers, put into the market more than 50 years ago, are recovered in the whole world shores, seas and seabed [5]. Due to its short lifespan and improperly management at end-of-life (EoL), packaging are those applications that most contribute to the MPP [6]. In particular, the results figured out by six of the beach clean-ups around the world show that out of almost 14 million items collected, five of the ten most found items are plastic packaging [6]. Among them, the single-use plastic packaging accounts for 50% of the marine litter [18,19]. It digs up correlation with the market trend of the last decades which sees a fourfold increase in the demand of conventional plastics from 1960 to date, accounting for 130 Mt in 2015 [11]. In Europe, the packaging sector registered a plastic demand of 20.1 Mt in 2018. About 40% is nowadays used to package food [2]. Looking at the waste management performance, only 42% of 17.8 Mt of plastic packaging waste (PPW) collected was sent for recycling while the remaining was landfilled (18.5%) and incinerated for energy recovery (39.5%) [2]. Many challenges affect the recyclability of packaging. Among them, the design largely contributes to underperform the material valorization of packaging in further cycles.

This paper presents the case of a small-medium enterprise that implemented an integrated decision-making process to rethink the packaging design, thus contributing to maximize efficiency at the EoL as well as in logistics and distribution operations. In this context, the work aims at highlighting that the development of a systemic and life cycle approach is crucial to reduce the environmental impacts of the packaging sector, especially when correlated to the urgency to prevent the use of plastic materials.

## 2. European Policy on Plastics Prevention and Packaging Reduction

Packaging manufacturers and users are under pressure because of the urgency to systemically rethink production, consumption, and disposal patterns. Many legislative measures have been introduced in the fields of plastics and plastic packaging to prevent problems in recycling infrastructure as well as natural environment. The *Circular Economy Action Plan (CEAP)* represents the starting point of a transitioning process towards more sustainable packaging system. In this framework, plastic materials have been considered as one of the five prioritizing materials [20,21]. The *Strategy for plastics in a Circular Economy* introduces the ambitious goal about having 100% reusable and/or recyclable packaging by 2030 [22]. The most striking *Directive on Singe-use Plastics (SUPs)* contains a multitude of actions aimed to ban and restrict the use of some plastic applications [23]. The *Packaging and Packaging Waste Directive* includes additional aspiring targets on plastic packaging waste collection and recycling [24,25]. While the first cycle of measures on circular economy invested on optimizing the recycling as option to add and maintain the intrinsic value of materials as long and better as possible in the value chain, the *new CEAP*, published by the Commission in 2020, prioritizes actions on reducing and reusing materials before recycling them [26]. Prevention covers a pivotal role in the *Green New Deal* as well [27]. Measures to reduce over-packaging are also highlighted in the new policy on sustainable products [26]. It follows that reduce and reuse strategies are the priorities of the current political agenda.

This commitment on the circularity and sustainability of packaging has stimulated the reaction of the converting industry that is putting effort in redesigning their products to prevent the use of plastic materials on one side and maximize the recyclability of products on another. Packaging issues emerge as the key elements of the sustainability agenda of manufacturing companies. A multitude of strategies, including sustainable resources supply, materials selection, recyclability and reusability systems are under exploration in Europe [28]. According to Bocken et al. [29], the following fundamental strategies towards the cycling of resources can be identified:

- Slowing resource loops through the design of long-life goods and product-life extension;
- Closing resource loops through the recycling.

In all the strategies, eco-design covers a fundamental role to speed up the plastic circularity. It takes more importance in one-way plastic packaging where the durability of the materials doesn't fit with the short life span of the applications [30]. The *Eco-design Working Plan 2016–2019'* published

by the European Commission in 2019 moves towards this direction by highlighting the necessity to further establish the scientific basis for developing corresponding quantitative and qualitative criteria related to material efficiency aspects [21].

## 3. Eco-Design in Packaging Applications

Eco-design, which consists in integrating environmental aspects at the preliminary stage of the design phase, represents the crucial phase to maximize the value of materials along the value chain [31]. By approaching designing systemically, considerations on the life-span, the functions and the externalities are managed not as a problem to be dealt with at the end of the production process or after the product has completed its useful life, but must be kept in mind from the beginning [32,33]. Through this anticipatory approach, many solutions come into light. According to Le Blevennec et al. [34], five main eco-design principles influence the various stages of the lifecycle of product:

- Design for sustainable sourcing;
- Design for optimized resource use;
- Design for environmentally sound and safe product use;
- Design for prolonged product use;
- Design for recycling.

In the field of recycling, many eco-design guidelines and tools as well as recyclability certifications have been established. Among them, PlasticsRecyclersEurope created the *RecyClass* tool based on three-steps process: Test for technology/product approval; eco-design guidelines as results of the lab protocols; recyclability self-assessment and expert-checked to get out the certification. Similar approach has been used by *Cyclos* in Germany and *Recoup* in UK. European Commission is also working on the *Eco-design toolbox* that look at all the products and materials, from the qualitative and quantitative point of view by integrating technical and technological aspects with environmental emission. Major investments have been done by KIDV that introduced metrics to simultaneously assess reusability, circularity, and environmental impacts of packaging over than the qualitative recyclability check. It follows that, even if reusability or recyclability principles are integrated in the eco-design phase, Life Cycle Assessment (LCA) studies are vitally important to compare different alternatives and validate any decision on the product sustainability [35]. It follows that, LCA demonstrates the potential sustainability of the supply and the consumption stages from the quantitative point of view, by estimating the effective reduction in the greenhouse gas (GHG) emission.

## 4. Packaging Sustainability in Food Applications

The diffusion of the culture of sustainability and circular economy increasingly pushes companies to adopt green strategies and integrate circular business models in the corporate agenda [36]. Considering social aspects, the corporate social responsibility is becoming a priority for companies at all levels, as well as traceability and communication. They offer the opportunity to build a strong relationship with the consumer, creating loyalty and new business opportunities, also maximizing the creation of sustainable value and the prevention in the generation of surpluses at the system level. It assumes higher relevance in the packaging industry since 3.970 billion units of packaging were consumed in Europe in 2018 [37]. The increasing awareness of consumers on environmental issues as well as the urgent need to deal with the challenge of the urban littering and the microplastics pollution on the marine ecosystem, have led the food industry to face many challenges. The most urgent are related to those elements hampering the recycling process. These elements are mainly identified in the bad design of packaging [38]. Multilayer packaging are commonly used in the food sector [39]. If multiple components with numerous materials protect sensitive food products, the concurrent presence of plastics with metals (like aluminum) or fibrous materials (like cardboard) cannot be easily detected by the near-infrared (NIR) technology and so, sorted and recycled. Moreover, the lack of

harmonization in the Green-dot system as well as the improper behavior of consumers in packaging collection strongly contributes to emphasize the problem [7]. To continue, challenges are not only technical/technological and social but also cultural: The change in lifestyle, especially in developed countries, has contributed to an increase in the demand for 'Ready to eat' packages thus increasing the possibility to litter them [40]. The work published by Ellen Mac Arthur Foundation demonstrated as fast-moving consumer goods (FMCGs) are getting more and more attention, accounting for 35% of material inputs and 75% of municipal solid waste [41]. In 2019 31.1% of the total amount of packaging sold are used for grocery, followed by beverage (26.1%), perishable food (17.3%), fruit and vegetable (5.8%), frozen food (3.6%) and meat (0.9%). The remains are used to package household and personal care (12.5%), and PET care (2.7%). A study published by McKinsey on the European packaging market reveals that the material composition is characterized by the presence of plastics (62%) and paper and board (14%), followed by glass (12%), metals (11%) and a minimum part of flexible foils (0.002%).

## 5. Case Study: Sustainable Packaging in the Frozen Food Industry

The packaging of the food products is functional to the containment, protection, preservation, and promotion of the product. Labels support the access to information for the costumers by capturing their attention. For these reasons it is important that the proposed message is homogeneous and coherent: the product of a supply chain that presents itself as sustainable should be contained and proposed within a packaging that is sustainable itself. It receives a quite big applications in organic foods and assumes greater relevance in the field of frozen foods where packaging must ensure good quality over the time. The increasing market for these products results in a global value of USD 291.3 billion in 2019, with a Compound Annual Growth Rate (CAGR) of around 5 billion in the period 2016–2020 [42]. However, the confectionary of frozen foods has received little attention in last years. The design of existing frozen food package is generally limited to ensure shelf life, to the detriment of recyclability. Frozen food packaging are largely made of multi-materials that are not recyclable [36,39]. According to Gutierrez et al., packaging can be considered sustainable when enables food safety, by preserving and improving the health and hygiene safety of food, but also theconservation, by limiting environmental impacts and promoting positive changes in the communities [43]. So, many efforts have been done by the key players of the sector to make more sustainable packaging, by integrating food safety and material valorization.

The present work summarizes the activities performed by "Cartotecnica Reggiani", a small Italian enterprise active in the design of innovative food packaging. The object of the eco-design application is a cardboard-based box, aimed at containing frozen food, typically herbs or ingredients for seasoning, which can be extracted from the refrigerator, dispensed from the box, sealed, and refrigerated again. The cycle can be repeated several times and, for this reason, the boxes of this kind of food includes a plastic sealing. A new patented product has been created starting from the idea to avoid both plastic utilization and multi-material packaging, by considering just the possibility to use recyclable paper and cardboard.

In particular, as shown in Figure 1, the common plastic cap (see Figure 1a), that is usually present in the traditional box used for packing frozen herbs, has been removed, with the main aim to remove single-use plastic components, guaranteeing opening and closing operations by introducing 100% paper cover that is perfectly integrated in the whole paper box (see Figure 1b).

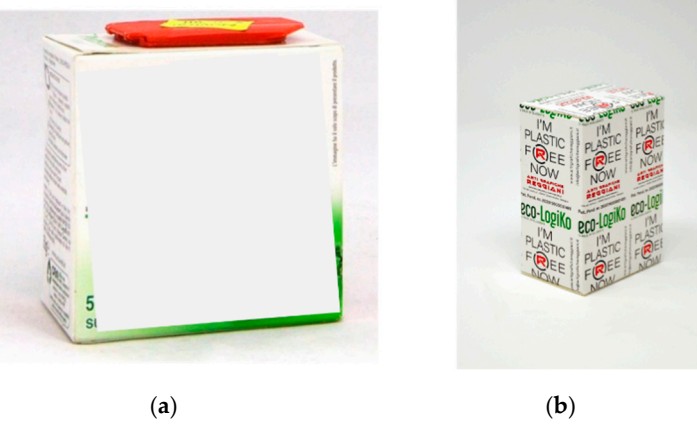

(a)      (b)

**Figure 1.** Frozen food packaging design: Standard and innovative items.

The shift from multi to mono-material and recyclable packaging has been supported by the implementation of an integrated decision-making process. Eco-design and LCA have been simultaneously used to test and validate the redesign process. Numerous studies, published in recent years, have applied the LCA methodology to food packaging [40,43–48]. An early example of scientific LCA work was presented by Kooijman in 1993, who studied the energy contribution of the packaging in the food chain [49]. More recent studies are instead focused on assessing the weight of the packaging to the total carbon footprint of the food-packaging combination [50]. However, as highlighted by Moberg et al., the packaging environmental issue should not be generalized, and it would be more appreciate to investigate the entire "packaging-product system" instead of the food-packaging alone [51]. In fact, if the existing research confirmed that recycling is more beneficial than incineration or landfilling, standardized eco-profile for different food packaging materials doesn't exist yet [46]. In fact, the implications of the functional unit, the system boundaries, the geographical collocation, the local waste governance, the transportation system, the data quality, and the allocation, can significantly vary the results [52]. It follows that each experimentation in redesign process should be massively followed by a validation test, represented by a comparative LCA study. The study demonstrates how packaging redesign contributes not only to make a more sustainable supply chain but also to maximize the logistic operations efficiency and energy savings, thus contributing to optimize the sustainability of the overall packaging system.

### 5.1. Materials and Methods

Due to the early stage of development of the product and in the perspective of future developments in terms of materials and possible application, eco-design requirements have been explored and a simplified LCA study has been performed in the R&D stage, with the aim of assessing the recyclability status and evaluating the environmental consequences of the innovative product in the system. For this reason, a comparative LCA has been outlined, and reiterated, to compare the sustainability status of the innovative food packaging designed and patented by the company, with the equivalent product already on the market. The overall impact has been assessed considering: First, the reduction of plastics within food product packaging; second, the reduction of the complexity of packaging to promote its recyclability; third, the optimization of the transport yield per unit of product, reducing the impact of transport; finally, the opportunity for costs containment.

Considering the existing eco-design guidelines, the elements hampering the recycling of general food packaging have been identified and replaced. Once the packaging has been defined as fully recyclable, the LCA study is performed. Following the international guidelines for LCA (ISO 14040 and followings) and the ILCD Handbook (REF), the procedure is carried out in different stages [53]. First, we have proceeded with the collection of all information related to types and quantities of raw materials, energy, and emissions into water, air, and soil (evaluation of materials and quantification of flows). Second, the identification of the key elements has been accomplished, such as the traditional

plastic "open and close" cap and the new closing method entirely made of cardboard. Third, direct and indirect impacts have been identified and calculated through software modeling, performed with the application of *Simapro 8* software. Finally, we performed an interpretation of the results obtained, with particular attention to the comparison between the solutions and identification of the environmental benefits.

In order to quantify the impact of the proposed solution, a reference stock corresponding to the average annual production of boxes for frozen food for a major costumer has been selected. In particular, boxes containing 100 g of frozen food product has been modeled, and 15,000,000 boxes was considered as the reference flow.

Details on the formulation of the conceptual model are reported in the following tables. Table 1 depicts the overall conceptual model, based on the comparison of the traditional solution with the innovative solution, Table 2 reports on the *Ecoinvent* database processes used within the *Simapro* calculation module and, finally, Table 3 completes considerations on logistics and the quantity of transportable product per unit of volume.

**Table 1.** Conceptual model.

| *Item* | Standard Packaging | Innovative Packaging | Modeling |
|---|---|---|---|
| FOOD-GRADE CARDBOARD | Use of functional materials for specific uses (e.g., for dry or frozen foods, powders, etc.) | Use of functional materials for specific uses (e.g., for dry or frozen foods, powders, etc.) | At present, the same food-grade cardboard is assumed. The next research steps may involve specific products, in order to obtain maximum Recyclability or compostability |
| FOOD-GRADE CARDBOARD— "PRESS-AND-TEAR" TAB | Present originally, it is cut and disposed of on first use. MATERIAL: It is made of the same material as the box. WEIGHT: the typical case for small frozen foods (e.g., garlic, onion, shallot, chopped parsley) was used as a model. A sample of objects available on the market placed the weight of the tabs between 0.17 and 0.22 g/100 g of product | Originally present, it is cut on first use and remains attached to the resealable Table It is made of the same material as the box. | As it does not constitute a distinctive element between the two solutions, it is not taken into consideration, except in terms of waste stream, which is considered negligible. |
| PLASTIC CAP | MATERIAL: typically made of PP (polypropylene) WEIGHT: the typical case for small frozen foods (e.g., garlic, onion, shallot, chopped parsley) was used as a model. A sample of objects available on the market placed the weight of the corks between 2.79 and 4.02 g/100 g of product | Absent | The environmental benefit of not disposing of plastic waste is considered. In particular, the target recycling percentages of 65% were applied (for the remaining 35% the incineration option was considered) |
| TRANSPORT | The standard product used for the construction of the conceptual model allows to stack 22 layers of bundles, for a total of 3080 cartons per pallet | The use of the Eco-Logic box allows for the optimization of palletization, adding 1 layer of boxes per pallet, with the same volume, for a total of 3220 boxes per pallet | The optimization of transport results in an increase of 4.5% of cartons transported, and, therefore, of product, per unit of volume |

**Table 2.** *Simapro* processes.

| Item | *Simapro* Process | Modeling |
|---|---|---|
| **PLASTIC CAP—PRODUCTION** | Polypropylene, granulate {RER}| production|APOS, U | The polypropylene production process, in the absence of specific information, was modelled using a process based on average European data. |
| **PLASTIC CAP—THERMOFORMING** | Thermoforming, with calendering {RER}| production|APOS, U | For the thermoforming process, designed to give the desired shape to the polypropylene, the process yield recommended by the *Ecoinvent* database was considered, i.e., 98%. |
| **TRASPORT** | Transport, freight, lorry, unspecified {RER}|transport, freight, lorry, all sizes, EURO5 to generic market for|APOS, U | In the absence of specific data, the European average distance of 200 km is considered, as per ILCD Handbook. |
| **PLASTIC CAP—FINAL DISPOSAL SCENARIO: INCINERATION** | Waste plastic, mixture {Europe without Switzerland}|treatment of waste plastic, mixture, municipal incineration|APOS, U | For the modelling of the end of life of the plastic cap, a given target of recycling of the same at 65% was considered (considering that PP belongs to packaging plastics for which a recycling chain is available). The residue (35%) was modelled as sent for incineration in a nearby plant. This plant was modelled using the *Ecoinvent* process based on average data from the European area (excluding Switzerland). |
| **PLASTIC CAP—FINAL DISPOSAL SCENARIO: RECYCLING** | PP (waste treatment) {GLO}|recycling of PP|APOS, U Il processo è stato completato da: – Electricity, medium voltage {RoW}| market for|APOS, U – Polypropylene, granulate {RER}| production|APOS, U | For the modelling of the end of life of the plastic cap, a given target of recycling of the same at 65% was considered (considering that PP belongs to packaging plastics for which a recycling chain is available). The residue (35%) was modelled as sent for incineration in a nearby plant. The recycling process was built on the basis of literature data, in terms of electricity consumption and material recovered. |

**Table 3.** Transportation process, details.

| Item | Quantity Modeled |
|---|---|
| Stock production (average year) | 15,000.000 |
| Cases per bundles | 10 |
| Bundles per layer | 14 |
| Layer height | 11.5 |
| Layers per pallet—standard packaging | 22 |
| Layers per pallet—innovative packaging | 23 |
| Cases per pallet—standard packaging | 3080 |
| Cases per pallet—innovative packaging | 3220 |

*SimaPro* is product system modeling and assessment software for LCA released for the first time in 1990, distributed worldwide by the developers, i.e., Pré Consultants, based in the Netherlands. The calculation method selected is "ILCD 2011 Midpoint + V1.10/EC-JRC Global, equal weighting". ILCD is the acronym for International Reference Life Cycle Data System and is the result of a project conducted by the Joint Research Center (JRC) of the European Commission which analyzed different life cycle impact assessment methodologies to reach consensus on the recommended method for each environmental issue, both at the midpoint (impact assessment) and the final one (damage assessment).

On the basis of the information provided by the company, the die-cutter, or the machine which, by means of blades arranged in the desired shape, engraves or cuts the chosen material, remains the

same in both cases. It is therefore reasonably possible to assume the same useful life expectancy for the cutting tool since no substantial changes have been made to the process or to the materials. In terms of LCA, it can therefore be said that the product function remains completely unchanged.

*5.2. Results*

As well-known, multi-material packaging are not recyclable because of the difficulties in detecting and then, separating materials that are part of the same item. It follows that the most critical component of the box has been recognized in the plastic cap. Encouraged by the corporate sustainability strategy based on the reduction of plastic materials supply, designers and engineers identified in paper and cardboard the alternative materials to use. Recyclability has been then checked by exploring the key elements promoting the sorting process in the material recovery facilities [54,55]. In particular, material composition, colors, sleeves, labels, and closures have been investigated (see Table 4).

**Table 4.** Key Eco-design requirements for recyclable packaging.

| REYCLABLE PACKAGING - 5 Key Requirements | | Changes Done |
|---|---|---|
| 1. MONO-MATERIAL PACKAGING<br>Avoid mixed materials or materials of the same type. If different materials have to be used, different densities are necessary. | ✓ | Plastic cap has been removed and substituted to paper cap, thus having 100% paper-based box. |
| 2. LITTLE COLORFUL PACKAGING<br>Minimise colour. If colour is necessary, avoid strong colours as much as possible. | ✓ | The box is basically white. |
| 3. MONO-COMPONENT PACKAGING (SLEVEES)<br>Avoid full sleeves. If a sleeve must be used, it should be easily removable and there should be clear instructions explaining how to detach it. | ✓ | Sleeves are not used since information are directly printed in the packaging surface. |
| 4. MONO-COMPONENT PACKAGING (CLOSURES)<br>Use easily separable closures and ideally recyclable. | ✓ | Plastic closure has been removed and substituted with paper cap that is an integral part of the main body. |
| 5. MONO-COMPONENT PACKAGING (LABELS)<br>Use small, easily removable labels that should cover no more than 60% of the product's surface area, as well as being easy to remove by using water-based glue. | ✓ | No labels have been used since information are directly printed in the packaging surface |

Considering the early stage of product development in which the LCA has been applied, the results are limited to the comparative scenario presented, based on the average annual production of boxes for a major retailer. In particular, results presented in the following Table (see Table 5) are related to the positive impacts triggered with the elimination of the plastic sealing, thus considered an environmental benefit, as it is completely substituted by a different methodology of processing the cardboard material composing the box itself. The second emerging environmental benefit is derived from the optimization of the transport process, with an increased number of boxes transported per unit volume.

**Table 5.** Calculated impacts, by process and impact category (ILCD 2011+).

| Impact Category | Unit | Transport | Plastic Sealing | Overall Environmental Benefit |
|---|---|---|---|---|
| Climate change | kg CO$_2$ eq | 5364.3354 | 144361.16 | 149725.49 |
| Ozone depletion | kg CFC-11 eq | 0.0009 | 0.0046 | 0.0055 |
| Human toxicity, non-cancer effects | CTUh | 0.0011 | 0.0241 | 0.0251 |
| Human toxicity, cancer effects | CTUh | 0.0002 | 0.0063 | 0.0066 |
| Particulate matter | kg PM2.5 eq | 2.2520 | 53.3628 | 55.6149 |
| Ionizing radiation HH | kBq U235 eq | 366.7918 | 20752.87 | 21119.66 |
| Ionizing radiation E (interim) | CTUe | 0.0023 | 0.0698 | 0.0722 |
| Photochemical ozone formation | kg NMVOC eq | 18.3464 | 316.0784 | 334.4249 |
| Acidification | molc H+ eq | 21.3207 | 513.5841 | 534.9048 |
| Terrestrial eutrophication | molc N eq | 61.9180 | 894.9881 | 956.9061 |
| Freshwater eutrophication | kg P eq | 0.5450 | 41.8917 | 42.4367 |
| Marine eutrophication | kg N eq | 5.6968 | 88.9898 | 94.6866 |
| Freshwater ecotoxicity | CTUe | 60709.85 | 1516473.03 | 1577182.88 |
| Land use | kg C deficit | 16218.03 | 83615.29 | 99833.32 |
| Water resource depletion | m$^3$ water eq | 0.0353 | 698.7861 | 698.8214 |
| Mineral, fossil & ren resource depletion | kg Sb eq | 0.5642 | 9.1386 | 9.7028 |

In order to capture the entity of the overall environmental benefit triggered by the innovative solution proposed and to support the company in the dissemination toward the value chain, the results obtained in terms of global warming potential reduction has been compared to the emission of a city car, travelling yearly for the reference distance of 20,000 km. Considering data available, the implementation of the innovative solution proposed on the average production for a major retailer would spare CO$_2$ emissions equivalent to 56 cars. Accounting, then, the solution proposed would not require any modification in the production process of the boxes, both in terms of materials and expected lifespan of the machinery, this can be considered a net environmental benefit, which, in perspective, could be extended with further applications of the same solution to other products.

## 6. Discussion

The study carried out made it possible to:

a. remove plastics by doing an alignment between the value proposition and the mission of the company;
b. make a fully recyclable packaging;
c. reduce the environmental impact at product and process as well as system level.

The study demonstrated the main direct and/or indirect environmental benefits deriving from the solution, which can be summarized in quantitative reduction of raw materials, elimination of disposable plastics, reduction of plastic waste quantities, better recyclability, storability and consequently optimization of transport, and more in general, promotion of the sustainability culture as highlighted within the *European Circular Economy Strategy*.

Even if such positive impacts cannot be detected by the existing tools, the innovative packaging also facilitates the proper waste collection and the reduction of the possibilities to contribute to MPP.

Based on the data collected, the information shared, the modeling carried out and the results obtained, it is clear that the proposed solution reduce the environmental impact at a systemic level, make it advisable for large-scale application.

## 7. Conclusions

The sustainability corporate agenda of a small-medium frozen food industry has led to rethinking the design of the packaging. An integrated decision-making process has been applied to simultaneously investigate, test, assess and validate the recyclability and sustainability of the item. In particular, eco-design and LCA have been applied to support the design of 100% paper-based box that is patented by the company. The new designed solution provides positive effects in each step of the value chain. As checked trough the Eco-design toolbox, the packaging can be considered fully recyclable. The LCA results demonstrate the higher environmental benefits in terms of raw material supply, manufacturing, and transportation. Moreover, additional key findings can be considered in the collection stage and in the after-use cycle. Specifically, the adopted solution sounds also effective in promoting the product sustainability. From the marketing perspective, the redesign has also the purpose to convey information to the consumer by capturing their attention. The message is homogeneous and coherent: The product, that is assumed to be related to a sustainable food chain, is contained within a packaging, which in turn is sustainable. Then, the new packaging is safe from the point of view of hygiene, health, and food safety. Finally, it has a limited environmental impact promoting an after-use economy and contributing to create added social value.

The proposed environmental life cycle assessment study can represent the indispensable starting point on which to base a subsequent *Environmental Product Declaration* (EPD). Developed in application of the UNI EN ISO 14025: 2010 standard (Environmental labels and declarations—Type III environmental declarations), the EPD is an excellent tool for communicating objective, comparable and credible information relating to the environmental performance of products. The EPD represents an important internationally recognized ecological certification and labeling, that will be reached by the analyzed product. In addition to the EPD, Life Cycle Costing (LCC) and Social Life Cycle Assessment (SLCA) can be applied to assess the triple bottom line of sustainability.

**Author Contributions:** Conceptualization, E.F.; methodology, E.F. and S.Z.; software, S.Z.; validation, S.Z.; formal analysis, A.B.; investigation, A.B. and S.Z.; resources, E.F.; data curation, S.Z.; writing—original draft preparation, E.F.; writing—review and editing, E.F. and A.B.; visualization, E.F.; supervision, A.B.; project administration, A.B.; funding acquisition, A.B. and S.Z. All authors have read and agreed to the published version of the manuscript.

**Funding:** This research received no external funding.

**Acknowledgments:** Authors acknowledge "Arti Grafiche Reggiani" for their fruitful and successful cooperation and for having provided all the data and information necessary for the development of the study.

**Conflicts of Interest:** The authors declare no conflict of interest.

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
