# Peer review of "Combining Eco-Design and LCA as Decision-Making Process to Prevent Plastics in Packaging Application"

_sustainability, doi:10.3390/su12229738_

Round 1
Reviewer 1 Report
The article discusses the important problem of introducing sustainable development and the circular economy. The case of a company that simultaneously introduced Eco-design and Life Cycle Assessment to design a new packaging for frozen food was considered. The authors found that the use of an integrated decision-making system has a positive effect on ecological aspects such as better recycling and lower energy consumption. Clearly written manuscript, well organized. The results of the study may be benefited to food packaging industry.
Author Response
Dear Reviewer, 
We are very thankful for your time and very good comments.
We hope that the paper can provide useful findings to the current research on the sustainability of packaging.
We review the entire work, especially for language and style. The new version is here attached.
With king regards,
The authors.
Reviewer 2 Report
The paper is interesting and the presented product is also interesting and promising. We nned howeverc some presentations of
simapro software at least in the supplementary materials. After that the paper is woth to be published in your journal
Author Response
Dear reviewer,
thank you for your comments.
We revised the paper to improve the English language and style.
We introduced a brief description of the Simparo software in the "Supplementary materials" section. Thank you for your suggestion.
Then, we optimized the reference list.
Thank you again for your time and availabilities.
With king regards,
The authors.
Reviewer 3 Report
Manuscript sustainability-1003926 is a well-written paper that brings the knowledge and highlights decisive strategies for the strong contribution to the reduction in the consumption of plastics and the prevention of marine/environmental pollution. The topic addressed herein is in the front line of research. The presented information is well organized. I have some suggestions for authors to improve further their study. These follow the text sequence:
-Abstract
Line 26 and elsewhere.'' Results showed how to use....that allowed...''.
-Introduction and elsewhere.
The sources, in my opinion, should be listed in the reference list.
-Discussion
Line 296. ''Even if these materials/contaminants cannot be detected...''.
Based on the aforementioned, I suggest a minor revision of the present study.
Author Response
Dear Reviewer, 
We are very thankful for your time and very good comments.
We really appreciate your insights. We took our time to rethink the paper and we have thoroughly revised it according to your suggestions. You will find below a list of replies for each of your comments:
- line 26: the linguistic revision has been done. Thank you for the suggestion.
- line 296 - 298: changes have been done. Thank you for your consideration.
- sources included in the paper as footnotes have been included as references. Sorry about that.
Finally, the authors reviewed the entire work, especially for language and style.
With king regards,
The authors.